# Normal Mode Analysis as a Routine Part of a Structural Investigation

**DOI:** 10.3390/molecules24183293

**Published:** 2019-09-10

**Authors:** Jacob A. Bauer, Jelena Pavlović, Vladena Bauerová-Hlinková

**Affiliations:** Institute of Molecular Biology, Slovak Academy of Sciences, Dúbravská cesta 21, 845 51 Bratislava, Slovakia

**Keywords:** normal mode analysis, X-ray crystallography, crystal structure, protein dynamics, elastic network model

## Abstract

Normal mode analysis (NMA) is a technique that can be used to describe the flexible states accessible to a protein about an equilibrium position. These states have been shown repeatedly to have functional significance. NMA is probably the least computationally expensive method for studying the dynamics of macromolecules, and advances in computer technology and algorithms for calculating normal modes over the last 20 years have made it nearly trivial for all but the largest systems. Despite this, it is still uncommon for NMA to be used as a component of the analysis of a structural study. In this review, we will describe NMA, outline its advantages and limitations, explain what can and cannot be learned from it, and address some criticisms and concerns that have been voiced about it. We will then review the most commonly used techniques for reducing the computational cost of this method and identify the web services making use of these methods. We will illustrate several of their possible uses with recent examples from the literature. We conclude by recommending that NMA become one of the standard tools employed in any structural study.

## 1. Introduction

This special issue contains many articles describing several of the exciting new developments that have occurred in X-ray crystallography in recent years. In this review, we would like, however, to call attention to an old, well-established, but underused technique—Normal Mode Analysis (NMA). NMA is a technique that can be used to describe the flexible states accessible to a protein about an equilibrium position. These states have been shown repeatedly to have functional significance, and there have been many reviews dedicated to the applications of NMA to the study of biological macromolecules [1,2,3,4,5,6,7,8,9]. At least one book has also been written on this topic [10]. NMA is probably the least computationally expensive method for studying the dynamics of macromolecules, and advances in computer technology and algorithms for calculating normal modes over the last 20 years have made it nearly trivial for all but the largest systems. Despite this, it is still uncommon for NMA to be used as a component of the analysis of a structural study. For example, the number of Web of Science citations (excluding reviews) for the most commonly used NMA software over the last 20 years is only 1757, while the PDB grew by more than 141,000 structures during the same time period. Even allowing that a single study typically produces multiple structures, this is still an enormous discrepancy.

What makes this situation even more surprising is that NMA in some form has either been used as or suggested to be used as a tool for aiding molecular replacement [11,12] or for model refinement [12,13,14,15,16,17,18] since at least the early 1990s. It has also been used to help fit models into low-resolution cryo-EM maps [19,20,21] and interpret small-angle X-ray scattering (SAXS) results [22,23]. While it is true that, in earlier years, the equipment and expertise needed for X-ray crystallography and NMA were rarely found together, thereby necessitating separate studies, this no longer needs to be the case. There is no reason why NMA should not become a standard part of the analysis of any newly determined structure. In this review, we will describe NMA, outline its advantages and limitations, explain what can and cannot be learned from it, and address some criticisms and concerns that have been voiced about it. We will then review the most commonly used techniques for reducing the computational cost of this method and identify the web services making use of these methods. Finally, we will review a few select examples from the literature where NMA was used as part of a structural study to answer questions of dynamics by way of illustration, and we conclude with our recommendations. It should be noted at the beginning that our goal is to show how NMA can be useful for a typical structural study rather than to review all the applications of NMA to X-ray crystallography or to highlight the most impressive applications of NMA.

## 2. Normal Mode Analysis

Normal mode analysis is a technique that can be used to describe the flexible states accessible to a protein or other molecule about an equilibrium position. It is based on the physics used to describe small oscillations, which can be found in any book on classical mechanics [24]. The idea is that, when an oscillating system at equilibrium, for example, a protein in an energy minimum conformation, is slightly perturbed, a restoring force acts to bring the perturbed system back to its equilibrium configuration (the minimum energy conformation in the case of protein structures). A system is defined to be in equilibrium or at the bottom of a potential minimum when the generalized forces acting on the system are equal to zero. At the minimum energy conformation, represented by the generalized coordinates $q^{0}$, the potential energy equation can be represented as a power series in *q*:(1)$$V\left(q\right)=V\left(q^{0}\right)+{\left(\frac{\partial V}{\partial q_{i}}\right)}^{0}\eta_{i}+\frac{1}{2}{\left(\frac{\partial^{2}V}{\partial q_{i}\partial q_{j}}\right)}^{0}\eta_{i}\eta_{j}+\ldots ,$$
where $q_{i}$ and $q_{j}$ represent the instantaneous configuration of components *i* and *j* and the deviation of component *i* from its equilibrium configuration is given by $\eta_{i}=q_{i}-q_{i}^{0}$. Superscripts of 0 indicate the equilibrium conformation. V(q) is the potential energy equation of the system and, for proteins, usually takes the form of one of the commonly used molecular dynamics force fields (an overview of these may be found in [25,26]). The first term in the series represents the minimum value of the potential and may be set to zero. The second term will be zero at any local minimum, so the potential can therefore be written
(2)$$V\left(q\right)=\frac{1}{2}{\left(\frac{\partial^{2}V}{\partial q_{i}\partial q_{j}}\right)}^{0}\eta_{i}\eta_{j}=\frac{1}{2}\eta_{i}V_{ij}\eta_{j},$$
where $V_{ij}$ is the Hessian matrix that contains the second derivatives of the potential with respect to the components of the system. The Hessian matrix contains information about how changes in the position of one component of the system are tied to changes in the others.

It is also necessary to consider the kinetic energy (*T*) of the system since we are interested in dynamics. For component *i*, this can be given by
(3)$$T\left(q\right)=\frac{1}{2}M\frac{d^{2}\eta_{i}}{dt^{2}},$$
where *M* is a diagonal matrix containing the mass of each particle. The entire equation of motion can be written as
(4)$$\frac{1}{2}M\frac{d^{2}\eta_{i}}{dt^{2}}+\frac{1}{2}\eta_{i}V_{ij}\eta_{j}=0.$$

One solution of this equation is the oscillatory equation
(5)$$\eta_{i}=a_{ik}\cos\left(\omega_{k}t+\delta_{k}\right),$$
where $a_{ik}$ is the amplitude of oscillation, $\omega_{k}$ is the frequency, and $\delta_{k}$ is a phase factor. By substituting this into Equation (Equation 4), the equation of motion can be rewritten into a standard eigenvalue equation:(6)VA=λA,
where the matrix *A* contains the $A_{k}$ eigenvectors of the Hessian matrix *V* and λ is a diagonal matrix containing the $\lambda_{k}$ eigenvalues. The $A_{k}$ eigenvectors are the normal mode vectors and describe in which direction and how far each particle in the system moves with respect to the others; the $\lambda_{k}$ eigenvalues equal the squares of the frequencies with which all particles involved in each mode vibrate. Each eigenvector describes in which direction each particle moves and how far it moves with respect to all the other particles in the system, but it does not provide absolute displacements, meaning that simple NMA cannot generally be used to provide the displacement amplitudes of a given normal mode [7].

Generally, the vibrational energy of the system is equally divided across all modes such that the average oscillation amplitude of mode *k* scales at $1/\omega_{k}$. Modes with higher frequencies, however, have energetically more expensive displacements than those with lower ones. Together, this means that the system experiences the greatest displacements along the lowest frequency or slowest modes: higher frequency modes typically describe rapid local motions, while low frequency modes describe slower collective ones or large-scale conformational changes. Assuming that the system is properly at equilibrium or in an energy minimum, the first six modes will have frequencies of 0. These represent collective translations and rotations of the whole molecule along or about the three Cartesian axes and do not change the internal potential energy of the system (these can be eliminated beforehand by imposing the conservation of linear and angular momenta on the system). The vibrational modes are orthogonal or normal to one another, meaning that they can move independently. The excitation of one mode does not trigger the motion of a different mode. The general motion of the system can be described by a superposition of all modes.

These normal modes yield analytical solutions to the equations of motion: for a given set of initial positions and velocities, NMA allows us to calculate where each atom of the system in question will be at any subsequent time subject to the small oscillation approximation. The system is assumed to fluctuate so little that it behaves for all practical purposes as a solid. Proteins have been experimentally observed to behave in this manner, but only at temperatures below 100–200 K [27,28,29]. Under these conditions, proteins, like other solids, can vibrate with specific modes of vibration, where each mode is characterized by a given frequency and a given atomic displacement pattern. Protein vibrations typically fall in the range from about 1–3500 cm${}^{-1}$. In the higher frequency modes, only a few directly bonded atoms have substantial displacements, but large parts of the structure often take part in the lowest frequency modes.

NMA has been used in chemistry since the 1950s where its utility was demonstrated by reproducing vibrational spectra [30,31,32]. It was first applied to a peptide in 1979 [33] before being used to study full proteins such as bovine pancreatic trypsin inhibitor (BPTI) [34,35,36], hexokinase [37], crambin [38], human lysozyme [38,39], ribonuclease [38], and myoglobin [40,41]. Extension to larger systems was, however, hampered by the computational expense of the method. More recently, thanks to both advances in computer equipment as well as new methods based on simpler models, it has become possible to examine larger structures, including the skeletal ryanodine receptor [42], virus capsids [43,44,45], Ca-ATPase [46], Myosin-II [46,47,48], F1-ATPase [49], GroEL [50,51,52], the ribosome [53,54,55], the C-ring of the bacterial flagellar motor [56], and the yeast nuclear pore complex [57].

### 2.1. The Elastic Network Model

Although NMA is less computationally expensive than molecular dynamics simulation, it is still not trivial for proteins containing many hundreds or thousands of residues. The first problem is that a given structure must be energy minimized before NMA to ensure that the starting conformation is in a true minimum with respect to the chosen force field. The minimization must proceed until machine precision is reached, typically below 0.001 kJ/mol-nm, which is much more computationally demanding than the minimizations normally employed for other tasks (these generally need to reach only 10–100 kJ/mol-nm). Frequently, the results of this process distort the structure to a greater or lesser extent, leading to NMA being carried out on a structure different from the initial one. The second problem, and the computationally limiting factor, is the diagonalization of the Hessian matrix. For classical NMA, all *N* atoms in a structure, normally including the hydrogen atoms, must be used, making the total matrix 3N×3N in size. For large proteins with thousands of atoms, this can become intractable very quickly. As a result, a number of coarse-grained approximate methods have been developed which overcome both of these limitations [4,7]. The most common and widely used of these methods is the elastic network model (ENM).

The general idea of ENMs, first put forward by Tirion [58], is to replace the complicated semi-empirical potentials used in standard NMA with a simple harmonic potential:(7)$$V\left(q\right)=\sum_{d_{ij}<R_{c}}C{\left(d_{ij}-d_{ij}^{0}\right)}^{2},$$
where $d_{ij}$ is the distance between atoms or nodes *i* and *j*, $d_{ij}^{0}$ is the distance in the initial structure, and *C* is a “stiffness” constant assumed to be the same for all *i*–*j* pairs. Note that, by design, the input configuration is assumed to be a minimum energy one, and thus energy minimization against a potential is unnecessary. $R_{c}$ in this equation refers to a cut-off radius and the given sum is restricted to all pairs less than this value. This value is somewhat arbitrary, but, in practice, values of between 7.0–8.0 Å are used based on the observed distances between non-bonded atoms in experimental structures [59,60] (methods with greater coarse-graining may increase this to 12.0 Å). Most often, only the C${}_{\alpha }$ atoms are used for these calculations because they are sufficient for studying the backbone motions, which is all that is needed to characterize the low-frequency normal modes. Figure 1 illustrates the application of an elastic network model to hen egg-white lysozyme.

A number of different ways of implementing ENMs have been developed. The simplest form is the Gaussian network model (GNM) developed by Bahar and co-workers [61,62], which replaces the 3N×3N Hessian matrix with an N×N Kirchoff or valency-adjacency matrix Γ. This matrix is defined in terms of a spring constant $\gamma_{ij}$, which is defined based on the assumption that the separation distance $|R_{i}-R_{j}|=R_{ij}$ between the *i*th and *j*th C${}_{\alpha }$ atoms in the protein follows a Gaussian distribution. The potential is given by
(8)$$V_{\mathrm{GNM}}=\frac{1}{2}\sum_{ij}\gamma_{ij}{\left(\Delta \vec{R_{ij}}\right)}^{2},$$
where $\Delta \vec{R_{ij}}$ is a vector expressing the fluctuations in distance between the *i*th and *j*th C${}_{\alpha }$ atoms. This model assumes that the fluctuations are isotropic and, consequently, no information about the three-dimensional directions of motion can be obtained; however, eigenvalue decomposition of Γ does allow the contribution of individual modes to the equilibrium dynamics, the relative displacement of residues along each mode axis, cross-correlation between residues in individual modes, and square displacement profiles to be calculated.

An extension to the GNM is the anisotropic network model (ANM), which is simply a coarse-grained NMA. Only the C${}_{\alpha }$ atoms are considered and the potential given in Equation (Equation 7) is applied to them. This is the form originally suggested by Tirion [58] and incorporated and developed into the Molecular Modeling Toolkit (MMTK) by Hinsen [63], and widely used in a number of other tools, some of which will be described below. It gives the same information as the GNM and also gives information on the directionality of the fluctuations. The mean-square fluctuations (*B*-factors) and cross-correlations it produces do not agree quite as well with experiment as GNM, however [10,64,65]. On the other hand, ANMs can be used to generate alternative conformations in the close neighborhood of the initial structure by deforming the structure along the lowest frequency modes [5]. This prompted Zheng [60] and Lin and Song [66] to develop models which combined the best features of both into a single method. The ability to deform structures can be and has been used for a number of purposes, most notably for molecular replacement [11,12], SAXS [22,23], and fitting to cryo-EM maps [19,20,21].

While coarse-graining does allow ENMs to be scaled to very large molecular structures, it does suffer from the drawback of losing detailed information on local movements. The rotating-translating blocks (RTB) model of Sanejouand [67], which is implemented in the ElNémo web service [68], was constructed as a generalized method to alleviate this. In this approach, the protein or other macromolecule is divided into $n_{\beta }$ blocks made up of one or a few residues connected by elastic springs. Next, it is assumed that a good approximation to the low-frequency normal modes can be made by making linear combinations of the local rotations and translations of these individual blocks. Consequently, a $3N\times 6n_{\beta }$ projection matrix *P* is constructed and used to build a projected Hessian matrix $H_{\beta }$:(9)$$H_{\beta }=P^{T}HP$$
and $H_{\beta }$ is diagonalized with $A_{\beta }^{T}H_{\beta }A_{\beta }=\Lambda_{\beta }$, where $A_{\beta }$ is the eigenvector matrix diagonalizing $H_{\beta }$ and $\Lambda_{\beta }$ is the corresponding eigenvalue matrix. The resulting eigenvectors can be projected back into the full 3N-dimensional space using $A_{P}=PA_{\beta }$, where $A_{P}$ is a $3N\times 6n_{\beta }$ matrix containing the $6n_{\beta }$ lowest-frequency approximate normal modes. The limitations of the method arise from how the blocks are selected: it does not reproduce internal motions within the blocks, so a great deal of information can be lost if flexible regions are assumed to be rigid.

### 2.2. Concerns and Limitations

Some concerns are commonly expressed about the utility of NMA in general and ENM in particular [3]. For NMA, the most fundamental question is whether the conformational transitions of real biomolecules are actually harmonic. A number of neutron scattering and NMR [34,69,70,71,72,73] experiments have shown that the large-amplitude, low-frequency motions normal modes are often supposed to describe are actually anharmonic. The anharmonicity arises from severe solvent damping and the roughness of the potential energy surface of the real protein (that is, there are a number of local minima lying on the potential energy surface on the approach to the global minimum). It should be noted, however, that the NMA approximation is only valid over short time-scales and for small amplitude vibrations (according to the principles of statistical mechanics [74], at 300 K, the expected amplitude of a typical thermally-driven oscillation will be very small, typically less than 1 Å), not the much longer-period vibrations generally measured in these studies. More importantly, it is widely thought that large amplitude anharmonic motions do begin as small amplitude ones on a short time scale, which is exactly where the approximations are valid.

The anharmonicity of the motions can arise from a number of sources. For example, a ligand-induced conformational change (where a ligand might include anything from a water molecule to a whole protein) with an amplitude much larger than that given by low-frequency normal modes might begin with a low-amplitude thermal vibration, but can then become an activated process driven by the energy released as a consequence of the binding of the ligand. This extra energy could then push the structure farther along the corresponding low-frequency modes than thermal fluctuations alone.

The characteristic motions described by NMA have frequently been validated using principal components analysis (PCA) of molecular dynamics simulations. These simulations are presently able to describe motions with timescales of up to a few microseconds for smaller proteins. The mathematical formulations behind PCA and NMA are similar: both involve solving a system of linear equations, and the matrix diagonalized in PCA, the covariance matrix, is actually the inverse of the Hessian matrix used in NMA. The covariance matrix is filled empirically by analyzing a number of snapshots from a MD trajectory rather than by calculating a derivative. In both cases, the result tells us what motions a given structure are in principle able to execute starting from a given conformation. A number of studies have attempted to validate NMA low-frequency modes using short- or long-time MD simulations [75,76,77]. These studies generally find that the low-frequency normal modes and the first few principal components overlap considerably, meaning that they do generally describe the same types of motions, although there is not always a one-to-one correspondence between a given mode and its corresponding PC: sometimes, the order is permuted and other times a given PC may be composed of elements from more than one mode.

More fundamentally, the low-frequency normal modes from either all-atom or coarse-grained NMA have been shown to correspond to functionally relevant motions in proteins [5,7] and conformational transitions have been shown to follow one or only a few normal modes [78,79,80]. Three classic examples include hemoglobin, GroEL, and the ribosome. The allosteric conformational changes in hemoglobin in moving between its tense (T) form and its relaxed (R) form closely followed the collective motions captured by the second-lowest frequency mode [81,82]. A single, though rather high-frequency mode of the GroEL chaperonin (mode 18), was shown to describe the conformational change between the closed and open states of the GroEL–GroES complex structure [3,50,83]. Finally, the two subunits of the ribosome follow a ratchet-like motion, which is described by the third-lowest mode of the system [53,84]. In general, the opening and closing of domains or subunits in many enzymes follow their low-frequency modes [85,86], and ligand binding cooperatively triggers collective movements or stabilizes those conformers either favored by or accessible to the open conformation [3,87]. Several more recent examples will be described below.

A concern which might be raised against ENMs in particular is that such coarse-graining does not capture the actual motion of an all-atom NMA. In reply, it should be noted that coarse-grained NMA methods are not intended to capture the micro-level motions of individual side-chains, but the low-frequency large-scale motions of the modes with the largest collectivity. Indeed, the lowest frequency modes of all-atom NMA have been shown to be largely unchanged by differences in particular local interactions, and seem to depend only on the overall shape of the macromolecule [83,88,89,90,91,92,93,94]. Indeed, this observation was one of the original inspirations for the initial development of ENMs. The lowest frequency modes identified by ENMs have been shown to be largely the same as those from all-atom NMA [77,79,95], and at least one study has found that a coarse-grained NMA reproduces the range of conformations found experimentally in the HIV-1 protease [96]. Likewise, increasing the level of detail has also been shown to increase the ability of these models to estimate the anisotropic displacement parameters determined from 83 high-resolution structures [97], and a study in which the coarse-graining was increased from N/2 to N/40 found that the lowest-frequency mode produced by a 1/40 level of coarse-graining still had a correlation of 0.73 with the lowest-frequency all atom mode [89].

This does, however, highlight one major limitation of any NMA: unless a given mutation makes a substantial change to the conformation of a protein, NMA will show that the dynamics are the same, even if the protein exhibits very different biochemical behavior. For example, in a recent MD study on the N-terminal domain of the human cardiac ryanodine receptor, we found that the dynamic behavior of three mutants differed notably from the wild-type using PCA, although the average structures were very similar (all RMSDs were below 1.5 Å) [98]. Normal mode analysis using both ElNémo [68] and ProDy [99,100] showed that the three lowest-frequency modes of all three mutants were very similar to those of the wild-type (the overlaps were more than 0.80 in all cases). It should be noted here that it has been shown, in the context of all-atom NMA that, when different conformations are used for calculating the normal modes, the pattern of atomic displacements (the eigenvectors) generally remained consistent, but the frequencies (the eigenvalues) varied significantly, sometimes even greatly enough to cause the rank order of the individual eigenvectors to change [7,101,102,103]. Thus, it is often necessary to consider a set of the lowest frequency normal modes rather than just the lowest individual ones.

### 2.3. Web-Based Tools Using ENMs

The coarse-grained NMA methods described above have been employed in a number of tools which are available online or can be downloaded for offline use. The most widely used of these are ElNémo [20,68], AD-ENM [104,105,106], NOMAD-Ref [107], oGNM [108] and its successors iGNM [109,110] and DynOmics [111], ANM 2.1 [112,113], HingeProt [114], MolMovDB [115,116], iMODS [117,118], the successor to DFprot [119], and WEBnm@ [120] (Table 1). All of these services make use of ENMs, though NOMAD-Ref also provides the possibility to do all-atom NMA for molecules of up to 5000 atoms. ElNémo, ANM 2.1, GNM, and HingeProt provide the source code for their underlying services for download. Other offline tools include Hinsen’s MMTK [63] (http://dirac.cnrs-orleans.fr/MMTK.html), and ProDy [99,100] (http://prody.csb.pitt.edu/). The different servers provide different services, which will be detailed below. Many of these methods have recently been reviewed by López-Blanco and Chacón [8].

ElNémo, one of the oldest NMA web servers [20,68], uses the RTB approximation and can therefore use all atoms in its normal mode calculations. The results provided include the degree of collectivity of the modes (defined as the fraction of residues affected by a particular mode), residue mean-squared displacement, distance fluctuation maps, and the correlation between the provided *B*-factors and the computed ones. If two structures of the same protein are provided in different conformations, overlap analysis can be used to identify the mode that contributes most to the given conformational change. It is also possible to generate alternative conformations of arbitrary amplitudes based on these modes for use in molecular replacement.

AD-ENM is also one of the oldest NMA servers [104,105,106]. In its present form, it can handle proteins, DNA, and RNA. Like ElNémo, it provides an option for determining the contribution of each normal mode to an observed conformational change, but it also has options for determining the effects on the overall structure that a conformational change to a subset of residues would have. That is, if new placements for only a few residues are known (or if even just the identity of these residues is known), then the server can predict how the rest of the structure will deform as a result. This feature was first used in a study by Zheng and Brooks [121] on the coupling between the local dynamics of the nucleotide binding sites of myosin and kinesin and the global dynamics of these proteins. In addition to ordinary NMA, AD-ENM also has a number of related tools, including DC-ENM [105] to generate alternative conformations of an input structure, PATH-ENM [106] for generating transition paths between two given macromolecular structures using a mixed elastic network model, iENM [122], which does the same but uses an interpolated elastic network model, and EMFF [21], which uses a modified elastic network model to fit an initial protein structure into a given cryo-EM map. A SAXS-based modeling tool has also been developed [22], but is not yet available for general use.

The NOMAD-Ref server [107] uses the RTB approach to compute all-atom normal mode analyses, but it also provides the opportunity to use GROMACS [123] to calculate a true all-atom NMA for molecules of up to 5000 atoms (about 260 residues since the H atoms must be explicitly included). Like ElNémo and AD-ENM, it can perform overlap analysis and create alternative structures. What makes it unique is that it provides tools for the refinement of structures produced by X-ray crystallography or cryo-EM (As of this writing, the NOMAD-Ref server has been offline for four months for upgrading).

DynOmics [111] is a portal for the ANM 2.1 [113], iGNM [109,110], and oGNM [108] services. It allows the user to either look up the results of an NMA on a structure from the PDB in the iGNM database or to carry out an analysis on an uploaded structure using both ANM and oGNM. A comprehensive set of results are provided. ANM provides molecular motions and animations. Full atomic structures for ANM-driven conformers can be generated and downloaded as well as trajectories in NMD files for viewing in VMD (Visual Molecular Dynamics), a popular MD viewing application [124]. The server also offers the possibility to incorporate environmental effects on the computed normal modes. This feature is especially well-developed for lipid bilayers. The effects of the environment on either the protein alone or the protein and the environment together can be examined. GNM provides the mean-square fluctuations of residues (*B*-factors) together with their correlation to the experimental ones, cross-correlations between residue fluctuations, an inter-residue contact map, and properties of the GNM mode spectrum. Domain separations, potential functional sites, and potential sensors and effectors can also be identified.

Like DynOmics, MolMovDB [115,116] is a portal to a number of different services, including a database of protein motions and a movie gallery [115,125], a morph server [126] for producing plausible pathways between two different conformations, and a number of hinge or rigid domain prediction tools [116].

Another service for predicting hinges, HingeProt [114], uses a combination of GNM and ANM to identify possible hinge residues in protein structures. iMODS [118], the successor to DFprot [119], carries out NMA using internal coordinates and calculates molecular deformability using conformal vector field theory [127]. It compares calculated and experimental *B*-factors, calculates the covariance matrix, and illustrates the elastic network using a two-dimensional matrix. WEBnm@ [120] uses Hinsen’s MMTK [63] to calculate its normal modes and provides atomic displacement analysis, visualizes the normal modes, determines which modes contribute to an observed conformational changes between two structures and plots the correlation matrix.

## 3. Applications

The utility of NMA for a typical structural study can best be seen by examining some of the structural studies that used it as part of the analysis. A search of PubMed for the terms “Crystal Structure” and “Normal Mode Analysis” in the title or abstract over the years 2000–2019 returned 55 papers, of which 38 described a crystal structure study that employed NMA as part of its analysis. Of these, three focused on more general biophysical properties, including the temperature dependence of coordinate shifts of the lysozyme structure [128], an improvement to the refinement of a low-resolution structure of the simian immunodeficiency virus coat protein gp120 [129], and the subtle conformational changes of a lysozyme crystal structure caused by altering its relative humidity [130]. There was also an unusual application where NMA was used to discover where the domain boundaries might lie in a protein where these were somewhat ambiguous [131]. The remaining studies could be loosely classified into two groups: the largest focused on categorizing the overall flexible motions of the protein, while the second concerned itself with how these motions might be correlated with ligand binding or catalytic activation.

### 3.1. Studies on Overall Flexible Motions

The greatest number of studies used NMA to characterize the general flexibility and domain movements of the protein under study [132,133,134,135,136,137,138,139,140,141,142,143,144,145,146,147,148]. The use of NMA in these studies runs from the simple to the sophisticated. Probably the simplest example is its application to the armadillo repeat domain (ARM) of the human adenomatous polyposis coli tumor suppressor protein [140]. In this study, a NMA was carried out on the ARM domains of this protein, β-catenin, plakophilin 1 and p120 catenin, and the lowest frequency vibrational mode was found to be similar for all four: a left-to-right swinging motion of the N-terminal part relative to their C-terminal part. This suggested that the ARM domain, otherwise thought to be quite rigid, does have some intrinsic flexibility.

In a more sophisticated use of this technique, Stanek et al. [147] used NMA in their comprehensive structural and biochemical study of *Aquifex aeolicus* Hfq, an RNA-binding protein involved in the post-translational regulation of mRNA expression and turnover. Hfq self-assembles into homohexameric rings, which combine into larger dodecameric assemblies in small amounts. NMA showed that the most flexible part of Hfq was not likely to be functionally important because it was not involved in any of the lowest-frequency normal modes. NMA also showed that the N-terminal regions of the hexamer have great flexibility in solution but become rigid when packed against each other in the dodecameric complex. Finally, they were able to account for the conformation of the dodecameric P1 crystal form using one of the low frequency normal modes (a slight rotation of the two rings with respect to one another).

While individual point mutations and variations in sequence that do not change the overall shape of a protein lead to normal modes that are generally indistinguishable from the wild-type, mutations which do alter the conformation can be used to identify differences in dynamics. For example, Dian et al. [141] were able to show that a 39-residue deletion of the nuclear export factor CRM1 was less rigid than the wild-type form, which was able to account for the mutant’s lower thermal stability. In another study, Roy et al. [148] used a 10 ns MD simulation, NMA, and fluorescence lifetime spectroscopy to show that the conformational change exhibited by a given loop in the I249T mutation of the human cathepsin K protein reflected actual changes in the dynamics of the protein caused by the mutation and was not a crystal packing artifact.

As noted above, one of the uses of NMA is to produce possible alternative conformations and to confirm observed ones [132,133,134,135,136,137,146,147]. Confirming observed ones can take the form of verifying that different conformers observed in different crystal structures reflect real solution conformations and not crystal packing effects (e.g., [133,148]). Alternatively, NMA can be used to show how a given conformational change occurred; that is, it can be used to show the preferred directions of flexibility, which, in turn, can suggest possible functional consequences for the processes being studied (e.g., [135,136]). An example of using NMA for producing alternative conformations is Sathiyamoorthy et al. [146], where NMA was used to predict alternative solution forms which were then identified using SAXS. Less ambitiously, Ng et al. [137] used NMA to account for the fact that an archaeal homologue of the Schwachman–Bodian–Diamond syndrome protein appeared to exist as a very strong dimer in the crystal form, but as a monomer in solution.

Finally, a particularly sophisticated application involved the use of a hybrid NMA-geometric simulation method [149] together with NMR to show that calexcitin, which appears to form a single compact unit in the crystal structure, is actually comprised of two separate, well-constrained domains with a considerable degree of relatively independent motion [142]. This degree of flexibility allows the protein to open up an interdomain cleft to an extent much greater than was apparent from the crystal structure alone. In addition, the NMA-geometric simulation of a C-terminal mutant mimicking phosphorylation showed that the mutant, and therefore the phosphorylated wild-type form, had a greater range of motion in the C-terminal area, which appears to be associated with a putative GTPase activity in this protein.

### 3.2. Correlations with Ligand Binding and Catalytic Activation

The second most common application is to examine how normal modes might be used to describe the conformational changes that are observed or that might occur during substrate binding, product release, or catalytic activation [150,151,152,153,154,155,156,157,158,159,160,161,162,163]. This is probably one of the oldest uses of NMA and was applied at an early date to lysozyme [38,39,164]. In this protein, the two domains bend towards each other following a motion that can be effectively described using a single normal mode of the lowest frequency. The final closed state is similar to the conformation of the ligand-bound closed state. A particularly good example of this application is to the Leu/Ile/Val binding protein LivJ, part of the *Escherichia coli* ABC transport system [150]. In this example, shown in Figure 2, the protein also has a two-domain structure with a central cleft similar to lysozyme. In the presence of a substrate, the cleft closes through a hinged rigid-body domain movement. The conformation of the ligand bound state can be very well approximated by the lowest frequency hinge-bending mode calculated based on the open structure.

A similar analysis was done on the *Salmonella typhimurium* phosphoglucomutase [156]. Open and partially closed structures were available for this protein, together with the fully closed form of a eukaryotic homologue. The first low-frequency normal mode could be correlated with the conformational changes bringing the enzyme from its open to its partially closed form. A combination of two modes was able to bring the distance of two catalytically important residues from 14.2 to 6.9 Å, which is within 0.7 Å of the distance of the corresponding residues in the closed homologue structure, thereby mimicking the closed form of the enzyme.

Bodra et al. [162] used NMA to join structural and kinetic results together to describe how conformational changes in the subdomains of dehydroascorbate reductase 2 from *Arabidopsis thaliana* could be linked to the binding of its glutathione cofactor (GSH) or to the release of its oxidized form (GSSG). This enzyme operates using a bi-uni-uni-uni ping-pong mechanism in which the GSH cofactor and the dehydroascorbate (DHA) substrate interact with a catalytic cysteine residue in separate, sequential binding steps. In the structures reported in this study, the α2 helix and its connecting loops are seen to be the most conformationally labile, and many of the residues involved in binding GSH can be found here, though not the catalytic cysteine. The lowest frequency collective motion described a twisting or breathing motion between the N-terminal thioredoxin-type domain and the C-terminal α-helical domain, which partially opens and closes the active-site cleft. The largest motion occurred around the α2 region and a solvent-exposed loop in the C-terminal domain which is not involved in substrate binding. A possible consequence of this motion is that the α2 dynamics might regulate the affinity of the protein for GSH or GSSG. For example, the presence of GSH might favor one conformation over another or α2 dynamics could rapidly release GSSG from the binding site to allow DHA to bind.

Finally, by following the conformational shifts of those parts of a protein containing catalytic residues, it is sometimes possible to answer questions about the protein’s catalytic activity even in the absence of a substrate. For example, Danyal et al. [161] prepared three independent mutations in the MoFe component of the Mo-dependent nitrogenase protein between the P cluster, which holds two iron-sulfur clusters, and the FeMo cofactor where N${}_{2}$ reduction actually takes place. Normally, the Fe protein, a separate component containing an iron-sulfur cluster, is needed to transfer electrons to the MoFe component for reduction to occur. The process goes as follows: after being reduced by flavodoxin or ferredoxin and binding 2 ATP molecules, the Fe protein binds to the FeMo protein, passes an electron, hydrolyzes its ATP, and leaves. The mutations in this study allowed this part of the enzyme to reduce a number of substrates, including H${}^{+}$ to H${}_{2}$, azide to ammonia, and hydrazine to ammonia, when supplied with electrons from an external polyaminocarboxylate-ligated Eu${}^{2+}$. N${}_{2}$ could not be reduced in this way, and only the Fe protein can be used to reduce N${}_{2}$, which suggests that the Fe protein plays a role other than simply donating electrons. NMA of an Fe–MoFe complex showed that motion of the Fe protein is correlated with a response inside the FeMo protein in the region between the P cluster and the MoFe cofactor, suggesting that there is a dynamic coupling between the Fe protein and the region of the MoFe protein lying between the P cluster and the MoFe cofactor. This could easily have consequences for intermolecular electron transfer and substrate reduction.

## 4. Conclusions

As we have shown above, normal mode analysis is a very useful technique for determining which conformational states are accessible to a given macromolecule. It can provide much of the same information given by more computationally expensive methods, such as molecular dynamics simulation, at only a fraction of the cost. It can be used to characterize the general flexibility and domain movements of a molecule, to produce possible alternative conformations and confirm observed ones, and to describe the conformational changes that occur or might occur during substrate binding, product release, or catalytic activation. We have illustrated each of these possibilities with recent examples from the literature. Moreover, aside from a general knowledge of protein structure and dynamics, very little specialized knowledge is needed to make use of the results of NMA, at least at their most basic level. NMA is also a very well-established technique with a number of easy to use tools available. For all of these reasons, we suggest that normal mode analysis should take its place as one of the standard tools of any structural study.

## Figures and Tables

**Figure 1 molecules-24-03293-f001:**
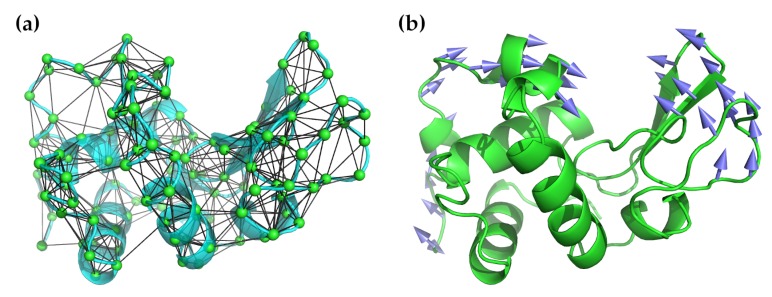
Illustration of the application ofNMA using an elastic network model to hen egg-white lysozyme (PDB ID 6lyz). (**a**) The lysozyme structure in cyan as a cartoon ribbon is overlaid with the elastic network model used for NMA. The C${}_{\alpha }$ atoms are shown as green spheres. All C${}_{\alpha }$ atoms within a radius of 7.3 Å are connected by black lines. (**b**) Lysozyme’s well-known hinge-bending motion is captured by the lowest-frequency non-zero mode. In this illustration, the arrows show the direction of motion, with longer arrows indicating greater motion.

**Figure 2 molecules-24-03293-f002:**
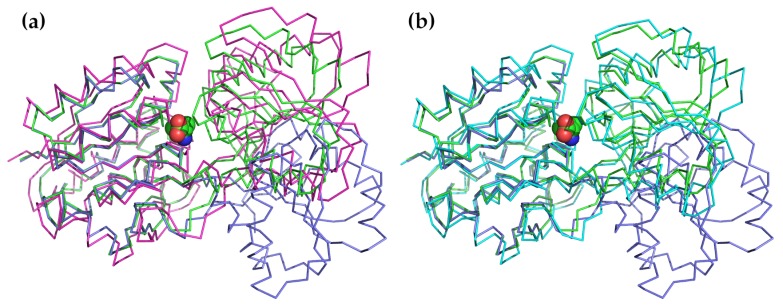
An illustration of how the conformational change between the open and substrate-bound forms of LivJ can be described using only a few low-frequency modes. The NMA was carried out on the open, unbound form of LivJ using the offline form of ElNémo [20,68]. In both panels, the unbound form is shown in purple, the substrate-bound form is shown in green, and the valine substrate is shown as Van der Waals spheres. (**a**) Most of the substrate-bound conformation can be reproduced by applying the lowest-frequency mode to the open form. The magenta structure shows the results of applying only the lowest-frequency mode to the open form of LivJ. The RMSD between this conformation and the substrate-bound form is 2.3 Å as compared to 6.5 Å between the open and substrate-bound forms. (**b**) The closed form can be even more closely approximated using a combination of five of the lowest-frequency modes (cyan). The RMSD between this form and the substrate-bound form is only 1.5 Å.

**Table 1 molecules-24-03293-t001:** The most commonly used NMA web services.

Service	url	Reference
ElNémo	http://www.sciences.univ-nantes.fr/elnemo/	[20,68]
AD-ENM	https://enm.lobos.nih.gov/index.html	[104,105,106]
NOMAD-Ref	http://lorentz.immstr.pasteur.fr/nomad-ref.php	[107]
oGNM	https://dyn.life.nthu.edu.tw/oGNM/oGNM.php	[108]
iGNM	http://gnm.csb.pitt.edu/index.php	[109,110]
DynOmics	http://gnm.csb.pitt.edu/index.php	[111]
ANM 2.1	http://anm.csb.pitt.edu/	[112,113]
HingeProt	http://www.prc.boun.edu.tr/appserv/prc/hingeprot/hingeprot.html	[114]
MolMovDB	http://molmovdb.org/	[115,116]
iMODS	http://imods.chaconlab.org/	[117,118]
WEBnm@	http://apps.cbu.uib.no/webnma/home	[120]

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
