# Peer review of "Normal Mode Analysis as a Routine Part of a Structural Investigation"

_molecules, 2019, doi:10.3390/molecules24183293_

Round 1

Reviewer 1 Report

This Review is timely and of general interest. Overall, the authors provide an up to date and comprehensive coverage of Normal Mode Analysis as one of the major simulation techniques, including the latest research developments in the field and of its applications in structural biology.

However some key issues should be addressed by the authors before this contribution should be accepted for publication in Molecules.

The effects of different force fields on NMA Section 1.2 Concerns and Limitations:                                             

 -The first paragraph should be thoroughly revisited and re-written: some of statements are incomplete (verbs are missing, etc.).             

 -The statement "PCA and NMA are related mathematically:" is IMHO not correct. The authors should rephrase as: "The mathematical formulations behind the PCA and NMA approaches are comparable:"     

-The authors should discuss and highlight the issue of NMA convergence radius vs rigid body refinement.                             

- For the sake of clarity, the reported examples (hemoglobin, GroEL and ribosome) should be  with  

Author Response

> Introduction: The effects of different force fields on NMA should be
> discussed.

I assume the reviewer means the effects of different force fields
(e.g. Amber, CHARMM, OPLS/AA) on the results of full, all atom NMA.
These were not discussed for two reasons. First, to do so would be to
go beyond the scope of this review, which is simply to illustrate how
useful NMA can be for analyzing a refined structure. Second, to the
best of our knowledge they are minimal. In particular, there does not
seem to have been any systematic study on the effects of these
different force fields on NMA in the way that systematic studies have
been done on the effects of these force fields on molecular dynamics
simulations. Much more has been done on the various methods of
carrying out approximate normal mode analysis, and these approximate
methods are normally linked to different ways of parameterizing the
molecule and, consequently, different force fields. These methods were
described in the manuscript.

> Section 1.2 Concerns and Limitations:
>
> -The first paragraph should be thoroughly revisited and re-written:
> some of statements are
> incomplete (verbs are missing, etc.).

No, none of the statements are incomplete and there are no missing
verbs.

> -The statement "PCA and NMA are related mathematically:" is IMHO
> not correct.
>
> The authors should rephrase as "The mathematical formulations behind
> the PCA and NMA approaches are comparable:"

They are related. How they are related is clearly stated in the
manuscript immediately following the colon.

Having said that, the suggested formulation doesn't really change the
meaning much at all, though I would prefer the word “similar” to
“comparable”.

> -The authors should discuss and highlight the issue of NMA
> convergence radius vs rigid body refinement.

Like the discussion of the effects of different forcefields on the
results of NMA, this is beyond the scope of this review.

> - For the sake of clarity, the reported examples (hemoglobin, GroEL
> and ribosome) should be split in three sub-sections.

These examples are found in one paragraph and each example has a
single sentence each. Their presence in this paragraph is simply to
illustrate that low-frequency normal modes can correspond to
functionally relevant motions. There is no reason to introduce three
one-sentence subsections here and to do so would very badly break the
flow of the overall discussion.

> - Please, consider the above suggestion also for the entire Section
> 2. Applications. The authors should clearly split the reported
> examples in defined sub-sections.

This was already done at the request of Reviewer 1.

> -The authors should avoid fine details i.e. line 211 " the
> second-lowest frequency mode”; line 214 "the third-lowest mode of
> the system". Non-experts in the field would hardly understand these
> details.

Actually, the initial draft was read by a non-expert who had no issues
at all with these details.

> - Line 241: "(the overlaps were more than 0.80 in all cases)".
> Please specify the units.

It's a fraction, equivalent to 8/10. There are no units.

> In general, this review would greatly benefit by the introduction of
> some graphical representation along the lines of those shown at
> http://thegrantlab.org/bio3d/tutorials/normal-mode-analysis

Two figures were added to illustrate an elastic network and an example
where a few normal modes can be used to reproduce a substrate bound
form.

> Section 1.3 Web-based tools using ENMs
>
> - The authors should consider the option to prepare a Table
> with the list of the most widely used web-based tools,
> instead of the current list within the text of the
> manuscript.

Fine, the table was added.

> Extensive editing of English language and style is required. The
> authors should avoid the use of colloquial English.
>
> -See some examples:
> Line 37 - "... and concerns that have been voiced about it."
>
> Line 204 - “Sometimes, the order is permuted and other times a given
> PC may be composed of elements from more than one mode.”

Our style is not any more colloquial than is common in scientific
writing. Moreover, such a style is typically much easier to read for
non-specialists than dense technical jargon. We also do not see what
is wrong in either of the two examples given by the reviewer (although
upon re-reading, it might, in fact, be better to join the sentence
given in line 204 to the immediately preceding one with a colon).

> The following reference should be included:
>
> Eric C Dykeman and Otto F Sankey “Normal mode analysis and
> applications in biological physics” J. Phys.: Condens. Matter 22
> (2010) 423202

Fine, added to the list of references.

Reviewer 2 Report

The manuscript entitled “Normal Mode Analysis as a Routine Part of a Structural Investigation” reviews recent uses of Normal Mode Analysis in the general framework of structural studies of biological macromolecules. Overall, the manuscript is well organized. There are however several minor concerns that need attention of the authors:

1) Pg 2, line53: “the minimum conformation” should read “the minimum energy conformation”.

2) Pg 2, lines 55-56: For the sake of clarity, it should be indicated that the superscripts of zero in equation (2) indicate the equilibrium conformation.

3) Pg 2, lines 64-66: Where does the equation (3) come from? Isn’t it missing a ½ factor? How is it derived from the classic T=1/2 mv2? The text states that it is kinetic energy from component i (a single particle?), however, M is a diagonal matrix containing the mass of each particle (for the whole system?).

4) Pg 3, lines 106-108: It should be stressed that the recent studies on larger structures are possible to not only more powerful computers, but also to new methods based on simpler models (such as ENM or RTB, that are described in the next section).

5) Pg 4, eq 8: Please tell the meaning of DeltaRij.

6) Pg 4, line 151: “macromolecular tool kit” should read “Molecular Modeling Toolkit”.

7) Pg 5, lines 168-169: Please define Vbeta, LAMBDAbeta and Abeta in the text.

8) Pg 5, line 177: Please rephrase this sentence.

9) Pg 6, line 252: In my hands, NOMAD-Ref server gives a Forbidden message for all tools, including NM calculations. If the server is no longer active, you should consider removing the reference to this service from the manuscript.

10) Pg 7, line 259: “of 5000 atoms and less” should read “of up to 5000 atoms”.

11) Pg 7, line 275-276: could you give an example where such a feature was used?

12) Pg 7, line 286: “5000 atoms (about 70 residues”: this cannot be right. That would make 71 atoms per residue. The biggest residue in a protein chain, Trp, has 24 atoms, including H’s.

13) Pg 7, lines 284-289: see comment 9.

14) Pg 8, section 2: This section is quite large. Please consider dividing it into two parts, one starting at line 328 dealing with the studies of flexibility, and another starting at line 375 devoted to the studies on the correlation flexibility-function.

15) Pg 10, line 422: There is an extra “the” at the end of the line.

16) Pg 11, line 449: “Anharmonic” should read “Anisotropic”; “macromolecular tool kit” should read “Molecular Modelling Toolkit”

17) Pg 11, line 462: “Pablo, C” should read “Chacón, P”.

18) Pg 14, line 596: “Through” should read “Thorough”.

Author Response

> 1) Pg 2, line53: “the minimum conformation” should read “the minimum energy conformation”.

Fixed

> 2) Pg 2, lines 55-56: For the sake of clarity, it should be indicated that the superscripts of zero in equation (2) indicate the equilibrium conformation.

Done

> 3) Pg 2, lines 64-66: Where does the equation (3) come from? Isn’t it missing a ½ factor? How is it derived from the classic T=1/2 mv2? The text states that it is kinetic energy from component i (a single particle?), however, M is a diagonal matrix containing the mass of each particle (for the whole system?).

Yes, there should be a 1/2 if I want to be consistent with the description of V I gave before. Also, since I am using generalized coordinates (as stated on p. 2 line 53), then I shouldn't be describing individual components, but the whole system. I was being somewhat sloppy in my notation. Fixed

> 4) Pg 3, lines 106-108: It should be stressed that the recent studies on larger structures are possible to not only more powerful computers, but also to new methods based on simpler models (such as ENM or RTB, that are described in the next section).

Okay

> 5) Pg 4, eq 8: Please tell the meaning of DeltaRij.

Done

> 6) Pg 4, line 151: “macromolecular tool kit” should read “Molecular Modeling Toolkit”.

Fixed

> 7) Pg 5, lines 168-169: Please define Vbeta, LAMBDAbeta and Abeta in the text.

Done (and I made a mistake: Vβ was supposed to be Aβ)

> 8) Pg 5, line 177: Please rephrase this sentence.

I have divided this sentence into two sentences, removed a couple of unnecessary words, and elaborated what was meant by the “inherent roughness of the real potential energy surface.” The result is longer, but hopefully easier to understand.

> 9) Pg 6, line 252: In my hands, NOMAD-Ref server gives a Forbidden message for all tools, including NM calculations. If the server is no longer active, you should consider removing the reference to this service from the manuscript.

Difficult. When we started writing this, the service was available but it was taken offline in April of this year for upgrades and hasn't been returned to service. It was claimed that the service was still available locally, but not to the outside world. The announcement of this is very hard to find and I have not been able to locate it again. I have sent an email to Marc Delarue asking him about the status of NOMAD-ref, but I have yet to hear back. If the server were no longer active, I imagine that the user-facing portal would have been removed as well, and I'm reluctant to remove what had been an established service that offered a number of services that are not available elsewhere. I'm going to leave it, but I will add a note that it has been offline for the last several months.

> 10) Pg 7, line 259: “of 5000 atoms and less” should read “of up to 5000 atoms”.

Okay

> 11) Pg 7, line 275-276: could you give an example where such a feature was used?

Done

> 12) Pg 7, line 286: “5000 atoms (about 70 residues”: this cannot be right. That would make 71 atoms per residue. The biggest residue in a protein chain, Trp, has 24 atoms, including H’s.

You're right. Given that the average number of atoms for an amino acid is about 19.20, 5000 atoms corresponds to about 260 residues.

> 13) Pg 7, lines 284-289: see comment 9.

Replied to above

> 14) Pg 8, section 2: This section is quite large. Please consider dividing it into two parts, one starting at line 328 dealing with the studies of flexibility, and another starting at line 375 devoted to the studies on the correlation flexibility-function.

Okay

> 15) Pg 10, line 422: There is an extra “the” at the end of the line.

Fixed

> 16) Pg 11, line 449: “Anharmonic” should read “Anisotropic”; “macromolecular tool kit” should read “Molecular Modelling Toolkit”

Fixed

> 17) Pg 11, line 462: “Pablo, C” should read “Chacón, P”.

Fixed

> 18) Pg 14, line 596: “Through” should read “Thorough”.

Fixed

Reviewer 3 Report

Bauer et al. provide an overview of the use of normal mode analysis (NMA) in protein structural biology. The review is organized into two main parts. The first part gives a theoretical introduction to NMA, introduces elastic network models (ENM), and summarizes potential limitations of NMA. The chapter closes with a detailed summary of web server implementations of NMA/ENM. The second part covers diverse applications of NMA of various complexity and gives a good overview of how the technique fits into the computational structural biology toolbox.

No recent review articles have covered the topic with the same scope, there is only partial thematic overlap with a recent review of Togashi and Flechsig (Int. J. Mol. Sci. 2018, 19, 3899), so the article is timely and relevant.

Overall, I was very impressed by the style of the manuscript, which leaves little room for improvement. I found the text easy to follow and very balanced when it comes to theoretical background and applications as well as older and more recent literature. The length of the manuscript is also appropriate, and the bibliography extensive. The manuscript is also very accessible for newcomers to the field.

I could only find a few typos that should be corrected before publication:

Line 312: "conformational change" instead of "chains"

Line 346: "variations IN sequence"

Line 390: "14.2 to 6.9 Å" instead of "14.2-6.9 Å"

Line 422: "the the"

Author Response

> Line 312: "conformational change" instead of "chains"

Fixed

> Line 346: "variations IN sequence"

Fixed

> Line 390: "14.2 to 6.9 Å" instead of "14.2-6.9 Å"

Okay

> Line 422: "the the"

Fixed

Round 2

Reviewer 1 Report

The authors have addressed most of the issues.

However, this reviewer is overall baffled by the authors' replies concerning the written English.

Overall, the manuscript would greatly benefit from a thorough proof read by a native English speaker to improve grammar.

Here some "typical" examples follow:

Section 1.2 - The Elastic network Model

Lines 170-172

Next, it is assumed that a good approximation to the low-frequency normal modes can be had by making linear combinations of the local rotations and translations of these individual blocks.

The sentence should be correctly phrased !!!

Section 1.2 - Concerns and limitations

Lines 183-185

A number of neutron scattering and NMR [34,69–73] experiments have shown that the large-amplitude, low-frequency motions normal modes are often supposed to describe are actually anharmonic. 

The sentence should be correctly phrased !!!

Lines 187-191

It should be noted, however, that the NMA approximation is only valid over short time-scales and for small amplitude vibrations (according to the principles of statistical mechanics [74], at 300 K, the expected amplitude of a typical thermally-driven oscillation will be very small, typically less than 1 Å), not the much longer-period vibrations generally measured in these studies.

The sentence should be correctly phrased !!!

Line 199

"farther" should read "further" 

Author Response

> However, this reviewer is overall baffled by the authors' replies
> concerning the written English.

> Overall, the manuscript would greatly benefit from a thorough proof
> read by a native English speaker to improve grammar.

The principal author, in turn, is baffled by the reviewer's comments
concerning the written English. The principal author IS a native
English speaker and cannot find any problems at all in the four
examples the reviewer chooses to highlight. He may allow that the
construction “an approximation can be had” is slightly less common
than the construction “an approximation can be made”, but the latter
is still perfectly legitimate. There is nothing whatsoever wrong with
the second and third examples. Finally, in the present context a
grammatical pedant might even go so far as to claim that “farther” is
actually the preferred word here (there is very little difference
between the two in any case).

At the prompting of one of the other authors, the principal author is
willing to change “can be had” to “can be made” in the first case, but
he sees no reason to make any other changes to the manuscript.